# From Development, Disease, and Decline: A Review of What Defines an Osteoclast Progenitor

**DOI:** 10.3390/ijms262110619

**Published:** 2025-10-31

**Authors:** Mitchell J. Shimak, Grant Kim, Ismael Y. Karkache, Elizabeth K. Vu, Emily Chavez, Joseph C. Manser, Emily Patterson, Archisha Basak, Keng Cha Vu, Samuel Mitchell, Jinsha Koroth, Elizabeth W. Bradley

**Affiliations:** 1Department of Orthopedic Surgery, Medical School, University of Minnesota, Minneapolis, MN 55455, USA; shima047@umn.edu (M.J.S.); kimx4465@umn.edu (G.K.); karka010@umn.edu (I.Y.K.); vu000085@umn.edu (E.K.V.); chave376@umn.edu (E.C.); manse044@umn.edu (J.C.M.); patt0454@umn.edu (E.P.); basak021@umn.edu (A.B.); vu000222@umn.edu (K.C.V.); mitc1048@umn.edu (S.M.); korot003@umn.edu (J.K.); 2Comparative and Molecular Biosciences, School of Veterinary Medicine, University of Minnesota, St. Paul, MN 55108, USA; 3Stem Cell Institute, Medical School, University of Minnesota, Minneapolis, MN 55455, USA

**Keywords:** RANKL, RANK, M-CSF, CSF1, CSF1R, bone remodeling, osteoporosis, surface marker, osteomorph, osteomac, bone healing, fracture, periodontitis, arthritis

## Abstract

Our understanding of the different developmental origins of osteoclast progenitors and their respective roles during homeostatic bone remodeling at different skeletal sites as well as their roles within bone regeneration and degenerative conditions is evolving. In this narrative review article, we summarize what is known about the developmental origins, anatomical sources, and markers of osteoclast progenitors. We touch on how osteoclast progenitors vary during different disease contexts, including periodontitis, rheumatoid arthritis, and osteoarthritis. In addition, we also characterize osteoclast progenitors that contribute to bone healing and define changes observed with advancing age. In this regard, we offer a critical review of gaps within our understanding and opportunities for future development within the field. Because of their diverse nature under different contexts, identifying and characterizing osteoclast progenitors may help to advance condition-specific therapies.

## 1. Introduction

The quintessential example of bone loss is osteoporosis, affecting 12.6% of US adults over 50 years old, with a higher prevalence among women at 19.6% [1]. While the cause of osteoporosis varies with its pathophysiological etiologies, primary osteoporosis is associated with enhanced bone resorption secondary to postmenopausal estrogen deficiency, leading to fragile bones [2]. Indeed, a 2019 US prospective cohort study by Ensrud et al. demonstrated that women older than 80 years diagnosed with osteoporosis had a 5-year hip fracture probability threefold higher than their non-osteoporotic counterparts [3]. Osteoporosis presents a significant burden to patients, their families, and health systems globally. Fragility hip fractures are associated with $10,075 worth of in-hospital costs, totaling $43,669 at 12 months [4]; furthermore, all-cause mortality of hip fractures is around 22% at 1 year [5]. Besides antiresorptive, pharmacological treatment, the American Academy of Orthopaedic Surgeons recommends that surgical intervention be taken as soon as medically feasible for patients who sustain hip fragility fractures [6]. Convalescence is further complicated by a risk of prosthetic joint loosening, with about 10% requiring re-operation within 15 years. Half of these revisions will take place secondary to periprosthetic osteolysis, causing aseptic loosening [7,8]. Beyond the hip, a 2019 meta-analysis of the global burden of bone fractures found there were 178 million new fractures with a 65.3% increase in years lived with disease since 1990 [9]. Elucidating the cellular mechanisms of bone homeostasis can lead to improved healthcare outcomes for patients with metabolic bone disease, fractures, and those undergoing orthopedic intervention involving implants.

A number of other pathologies, including rheumatoid arthritis (RA) and periodontitis, are also characterized by bone loss. During RA, pro-inflammatory cytokines such as TNF-α, IL-1β, and RANKL, produced by activated T-cells and synovial fibroblasts, directly stimulate bone resorption [10]. This is exacerbated by auto-antibodies found in RA [11], leading to focal bone erosions and periarticular osteopenia. Inflammatory factors of RA also impair bone repair, furthering net bone losses [11]. This all contributes heavily to the long-term sequelae of RA, including progressive joint destruction, deformities, and increased risk of fragility fractures, causing physical disability and significant decreases in quality of life [12]. RA is also a well-established risk factor for osteoporosis, tying it to all of the burdens listed above. A similar mechanism is seen in periodontal disease, except in this case, bacterial dysbiosis increases pro-inflammatory cytokine levels and bone resorption by triggering the host immune response [13]. In contrast to RA, bone loss during periodontal disease tends to be more local, characterized by alveolar bone loss which leads to edentulism [13].

Bone remodeling is facilitated through a continuous cycle of resorption and formation, which becomes unbalanced within disease contexts. This highly coordinated cycle relies on the balance between multiple cellular participants within the bone remodeling unit including: osteoclasts, osteoblasts and osteocytes (Figure 1) [14]. Osteoclasts are multinucleated cells derived from osteoclast progenitors of the monocyte/macrophage lineage. Osteoclasts function to resorb and degrade bone at sites of damage during bone remodeling [15]. These monocyte/macrophage cells fuse together in response to cytokine and receptor stimuli to form multinucleated osteoclasts [16,17]. CSF1-CSF1R, IL-34-CSF1R, and RANK-RANKL cytokine-receptor bindings provide signals for proliferation of precursors and stimulate the differentiation and resorptive functions. These cytokines also aid the survivability and maintenance of pre-osteoclasts and mature osteoclasts [18]. Osteoblasts are involved in various processes including the production of bone matrix proteins, mineralization of bone, and the expression of osteoclastogenic factors [19]. A subset of osteoblasts will undergo further differentiation and become bone-encased osteocytes [20].

There are five key phases involved during the bone remodeling process; these include (1) activation, (2) resorption, (3) reversal, (4) formation, and (5) termination. During activation, either mechanical strain or hormone signaling initiate bone remodeling [14]. When mechanical strain is put on the bone, osteocytes are responsible for sensing changes and using various dendritic processes to signal other cells to initiate bone remodeling [21]. Resorption ensues after activation and is accomplished by mature osteoclasts that developed from the precursors recruited during activation [21]. Following resorption, the reversal phase couples bone resorption to formation through the creation of an osteogenic environment at remodeling sites. The reversal phase begins when osteogenic signals are released from osteoclasts. These signals reach nearby cells, including bone marrow and bone surface cells. These bone lining cells on quiescent bone surfaces retract in order to allow the osteoclasts to have access to the bone matrix. Other cells termed reversal cells cover around 80% of the eroded surface [22]. There is a lack of knowledge about how the cells included in the reversal phase operate and the origins of osteoprogenitors [22]. This is a critical area of research, as development of biomaterials supporting bone formation could be greatly improved. The reversal phase is followed by formation, the longest phase of bone remodeling, which begins with the migration of osteoblasts to the resorption lacunae, stimulated by Wnt signaling [14]. Then, osteoblasts mediate osteogenesis. Following mineralization, osteoblast cells undergo apoptosis or change their function by becoming osteocytes embedded in the bone or lining cells that lay flat in quiescence. This marks termination, the last stage of bone remodeling [21].

In addition to hematopoietic progenitors, osteoclasts that facilitate bone resorption can be derived from other sources of progenitors. This includes embryonic erythromyeloid progenitors (EMPs) and potentially other non-canonical osteoclast progenitors such as osteomorphs and osteomacs. Osteomorphs are a newly discovered osteoclast lineage cell type in mice, which have a significant role in bone resorption. A study by McDonald et al. demonstrated that multinucleated osteoclasts could undergo fission to form smaller, motile daughter cells, termed osteomorphs [23]. Osteomorphs can circulate in blood and can also re-fuse to form osteoclasts, an alternate fate to apoptosis. In contrast, osteomacs are a subset of bone-resident macrophages that have been discovered as an important regulator of bone remodeling along with osteoblasts, osteoclasts, and osteocytes. Osteomacs constitute one-sixth of all bone marrow cells forming canopy-like structures within bone remodeling sites in the calvarium to support osteoblast survival, differentiation, and mineralization [24]. Studies suggest that they may influence osteoblast-driven mineralization and, under pathological conditions, act as a precursor for multinucleated giant cells [25,26,27]. This highlights the plasticity and the potential role of osteomacs in coupling osteoblast and osteoclast activity.

Traditionally, osteoclasts were thought to arise solely from hematopoietic stem cell-derived progenitors, but our understanding of the different developmental sources and their respective role in normal physiology, site-specific remodeling, and pathology is evolving. Because there are potentially unique osteoclast progenitors that orchestrate bone resorption/remodeling under different contexts, identifying and characterizing these cells is key to providing tailored therapies.

## 2. Sources of Osteoclast Progenitors

A growing body of research supports that there are multiple sources of osteoclast progenitors and that these pools vary across the lifespan and during different disease states (see Figure 2 for an overview). These osteoclast progenitors include EMPs and hematopoietic stem cell (HSC)-derived precursors. Osteoclast progenitors can arise from bone marrow-resident cells, mobilized splenic cells, as well as circulating peripheral blood monocytes. There is also budding evidence to support so-called non-canonical osteoclast progenitors, including osteomorphs and osteomacs.

### 2.1. Developmental Origins of Osteoclast Progenitors

Hematopoietic stem cells give rise to osteoclast progenitor cells. Bone marrow transplantation studies that demonstrate functional recovery of osteopetrosis in humans support the HSC origins of osteoclast progenitors [28,29,30]. Further studies in mice confirmed that osteoclast progenitors derived from HSCs originate from the bone marrow and spleen [31,32,33].

More recent evidence demonstrates that osteoclast progenitors also derive from erythromyeloid progenitors. Osteoclasts derived from embryonic EMPs seed neonatal bone, enabling marrow cavity formation and colonization of the bone marrow by hematopoietic stem cells and definitive hematopoiesis [34,35,36,37,38]. Lineage tracing studies in mice suggest that both early EMPs arising from the yolk-sac and late EMPs derived from the fetal liver serve as osteoclast progenitors (5). These EMP-derived progenitors give rise to osteoclast populations that persist into postnatal life and participate in both physiological remodeling and bone repair after injury [34,39]. EMP differentiation is independent of postnatal HSCs, but evidence suggests that EMP-derived osteoclasts may fuse with HSC-derived progenitors postnatally [34]. While there is much evidence to suggest that EMPs contribute to osteoclasts in pre-clinical models, existence of EMPs of embryonic origin in humans and their ability to form osteoclasts during embryonic development is unknown. In other organ systems, induction of embryonic phenotypes can rejuvenate tissue [40,41,42]. It is therefore tempting to speculate that induction or delivery of EMPs could attenuate pathological bone loss by rejuvenating bone-resorbing osteoclasts.

### 2.2. Bone Marrow-Resident OCPs

Bone marrow-resident cells constitute the dominant source of osteoclast progenitors under homeostatic conditions. These populations often support macrophage and dendritic cell differentiation as well [43]. Early fractionation studies in mice suggested the existence of multiple osteoclast precursor populations within the bone marrow [44,45,46,47], but this work did not identify direct progenitors or rule out macrophage differentiation. Other studies demonstrate that bone marrow-resident subsets define the predominant source of osteoclasts for steady-state remodeling, forming functional multinucleated osteoclasts dependent on mesenchymal stem cell interactions using murine models [39,48]. These marrow-derived cells reside within specialized stromal niches and are regulated by local mesenchymal stem cells (MSCs) and signaling molecules such as RANKL and CSF1, with Adipoq^+^ cells within the bone marrow potentially serving as the predominant source of CSF1 and RANKL [49,50,51,52]. Although most of these studies were performed in mice, they clearly document that bone marrow-resident progenitors contribute to osteoclastogenesis.

### 2.3. OCPs Within the Circulation

Under pathological conditions, such as inflammatory arthritis, bone metastasis, or advanced age, circulating peripheral blood monocytes are increasingly recognized as key contributors to dysregulated osteoclast activity. These cells, including CD14^+^ and CCR2^high^ subsets, are recruited to inflammatory bone environments via chemokine-mediated mechanisms, notably the CCL2/CCR2 axis [53,54,55,56]. Experimental blockade of this signaling axis reduces pathological osteoclastogenesis and bone resorption in mouse models of arthritis [53]. Further evidence from both human and murine systems confirms that peripheral blood-derived precursors can differentiate into functional osteoclasts in the absence of stromal support, highlighting their autonomously osteoclastogenic potential in inflammation. In addition to peripheral blood monocytes, myeloid-derived suppressor cells (MDSCs) can also function as a source of osteoclast progenitors in the context of cancer-induced osteolysis [57]. Intravital imaging studies likewise demonstrate that osteoclast precursors traffic through the bone vasculature, suggesting that cells from the circulation can participate in bone remodeling [58].

Circulating osteoclast progenitors can also originate from mobilized splenic cells. In vitro studies demonstrate that cells isolated from the spleen can support osteoclastogenesis [59,60]. Moreover, osteopetrotic mice lacking bone marrow-resident OCPs acquire osteoclasts with age in an IL-34-dependent mechanism, which was abrogated via splenectomy [61]. Using photoconversion-based cell tracking, marked cells were shown to traffic from the spleen to form labeled osteoclasts lining long bone surfaces [62]. Lineage tracing studies likewise demonstrate that the spleen is a source of osteoclast progenitors [36]. Combined, these studies demonstrate that peripheral osteoclast progenitors arise both from bone marrow-dependent sources as well as cells mobilized from the spleen.

### 2.4. Non-Canonical Osteoclast Progenitors

Two putative sources of OCPs include osteomorphs and osteomacs. Both osteomorphs and osteomacs have been experimentally characterized as non-canonical osteoclast progenitor populations that influence bone remodeling. In vitro evidence suggested that osteoclasts not only form via fusion, but can undergo fission [63]. This process can be induced by the soluble RANKL decoy receptor, Osteoprotegerin (OPG, TNFRSF11B, TR1, OCIF, PDB5) [23]. Studies utilizing in vivo fate-mapping and photoconversion labeling demonstrated that multinucleated osteoclasts undergo fission in vivo leading to osteomorph formation [23]. These cells rapidly re-fuse to restore bone resorption in response to RANKL [23]. Moreover, the osteomorph pool size quantitatively dictated the timing and magnitude of resorption “rebound” after anti-RANKL therapy withdrawal, establishing a direct functional link to bone remodeling outcomes. In contrast, the direct effect of osteomorphs on bone formation (osteoblast activity/mineralization) remains unexplored. Although the existence of osteomorphs may explain rapid bone loss associated with discontinuation of anti-RANKL therapy [23], evidence supporting osteomorphs within humans is also lacking.

Osteomacs are bone marrow-resident macrophages that support bone formation. A few key studies support that osteomacs display context-dependent plasticity to support osteoclastogenesis and promote coupling between resorption and formation [64,65,66]; however, this is under strong RANKL/M-CSF concentrations. Post-menopausal bone loss (i.e., ovariectomy models) shows increased osteomac numbers tightly associated with areas of heightened bone resorption and decreased formation [67], but direct osteomac-to-osteoclast differentiation remains limited to inference or rare events. Clear in vivo fate-mapping of osteomac-to-osteoclast conversion is needed to provide definitive evidence that osteomacs serve as non-canonical osteoclast progenitor pools. In contrast, in vivo evidence in mice and humans suggests that the major role of osteomacs is to support coupling rather than bulk replenishment of differentiating osteoclasts [24,66,68].

Collectively, these studies converge on a model in which distinct osteoclast progenitor sources are differentially engaged depending on developmental timing and disease context, each regulated by unique molecular and environmental cues. Emerging work suggests that embryonic progenitors possess persistent functional roles into adulthood [36] while circulating monocytes become dominant under inflammatory states [53], and that bone marrow-resident progenitors maintain bone turnover under baseline conditions [39,45,48]. However, direct comparative analyses of molecular profiles and functional outcomes across all three progenitor populations remain limited, particularly under pathological conditions. Continued investigation is needed to elucidate the regulatory mechanisms that specify osteoclast differentiation from these distinct progenitor compartments.

## 3. Defining Surface Markers of Osteoclast Progenitors

Osteoclast progenitors are cells that directly fuse to form multinucleated, bone resorbing cells. Osteoclast progenitor cells must respond to RANKL and M-CSF, but unfortunately the receptors for these cytokines, RANK and CD115 (Csf1r, c-fms), are expressed by other types of bone marrow resident cells as well as cells within the periphery. CD115 is broadly expressed by cells within the myeloid lineage, including osteoclasts, dendritic cells, as well as monocytes and macrophages, including tissue-resident cells as well as those within the circulation. RANK is mainly expressed by osteoclast precursors, mature osteoclasts, and immune cells such as DCs, macrophages, T cells, and microglia [69]. Expression is also detected in some epithelial cells in the small intestine [70], mammary glands [71], and some cancer cell types [72]. As such, additional surface markers are needed to define osteoclast progenitors. While many groups have identified osteoclast progenitors surface markers, a definitive characterization is still lacking. This may be because these cells are highly plastic and/or that multiple osteoclast precursors are present within different physiological and disease contexts. Below we discuss how several surface markers (Figure 3) have been used alone or in conjunction with others to isolate cells with osteoclastogenic potential from different anatomical sites and/or disease states.

CD11B/ITGAM (CDR3A, MO1A, MAC-1, MAC1A, SELB6) encodes the integrin alpha protein expressed by immune cells, including myeloid and lymphoid cells, as well and microglia within the brain and endothelial cells. ITGAM helps to facilitate leukocyte migration and infiltration. Because of its expression by monocyte/macrophage cells, CD11B was used early as a surface marker to enrich osteoclast progenitor populations. Both CD11B^+^ and CD11B^−/low^ populations, in combination with other surface markers, can be used to isolate bone marrow-resident and/or circulating osteoclastogenic cells from mice [45,73,74,75,76,77,78,79]. CD11B^+^ peripheral cells were mostly identified in studies concerned with inflammatory bone loss, including TNF-α-driven bone loss or within models of experimentally induced rheumatoid arthritis [74,76,78,79]. In contrast, CD11B^−/low^ bone marrow osteoclast progenitor cells were identified in homeostatic conditions. Most human studies use human blood to identify osteoclast progenitors and are most frequently found in the CD11B^+^CD14^+^ monocyte pool in adults [78,80]. In addition, studies using human cells of embryonic origin have identified CD11B as a marker for osteoclastogenic progenitors [43,47]. Overall, CD11B can be used in combination with additional markers (e.g., c-kit, CD115, Ly6c, lineage exclusion) to isolate osteoclast progenitors, but positive expression seems to delineate between OCPs involved in pathological versus physiological bone remodeling, respectively.

CSF1R (Fms, c-fms, Fim2, Csfmr, Cd115, M-CSFR) is the receptor for CSF1/M-CSF and IL-34 and is expressed by monocyte/macrophage cells with osteoclastogenic potential. CSF1R mediates the activation of multiple downstream kinases, including MEK/ERK, PI3K/Akt, JAK/STAT, and PLC. CD115-based positive selection identifies osteoclast progenitors in mice that form TRAP-positive multinucleated osteoclasts in vitro [81,82,83,84]; however, CD115 marks multiple hematopoietic lineage cells. Given this, most studies utilize CD155 positive selection in conjunction with additional surface markers to enrich osteoclast progenitors.

CD45R (B220, Cd45, L-CA, Ly-5, Lyt-4, Ptprc, T200) is broadly expressed by hematopoietic cells, but may be a defining surface marker to isolate the osteoclast progenitor pool. CD45 is a phosphatase that regulates activation of Src kinases. Early studies suggested that CD45R marked bone marrow-resident cells that formed osteoclasts in vitro in response to CSF1 and RANKL [73]. Moreover, the authors demonstrated that CD45R^+^ bone marrow-resident cells, but not CD45R^−^ cells, expanded following ovariectomy [73]. In contrast, later studies by this group demonstrated that highly purified CD45R^+^ cells did not support osteoclastogenesis [45]. Instead, CD115^+^CD117^+^ (Csf1r^+^C-kit^+^) bone marrow resident cells were identified, but the authors noted that CD117 was downregulated during osteoclast differentiation, indicating that these cells may not be direct osteoclast progenitors [45]. Further experiments demonstrated that FACS-sorted CD11B^−/low^CD45R^−^CD3^−^CD115^high^Cd117^+^ murine bone marrow cells contained the highest osteoclastogenic potential. This was corroborated by a study demonstrating that murine bone marrow CD45^−^CD3^−^CD11B^−/low^Ly6c^high^ cells demonstrated high osteoclastogenic potential that was exacerbated when cells were derived from animals with experimentally induced rheumatoid arthritis [74].

Other surface markers have likewise been used to isolate purified osteoclast progenitors. Peripheral monocytes are characterized by high surface expression of CD14 in the periphery in humans, with Ly6C serving as the mouse equivalent. As such, CD14 also marks peripheral osteoclast progenitors. Multiple rigorous studies in mice utilize c-Kit (CD117) in combination with other surface markers as a prospective positive selection strategy to isolate osteoclast progenitor populations from bone marrow and peripheral sources in mice and humans [47,85,86,87]. The chemokine receptor CX3CR1 is also a marker of osteoclast progenitors. Evidence using lineage tracing models demonstrates that CX3CR1 marks a pool of osteoclast progenitors in mice [37,39]. Although CX3CR1 marks osteoclast progenitors, several studies also demonstrate that these cells have multilineage potential and can generate macrophage and dendritic cell lineages in addition to osteoclasts.

Given that the evidence for both bone marrow-resident and circulating osteoclast progenitors in different developmental and disease-specific settings, surface markers used for the isolation of these cells may be context dependent. Although direct evidence is lacking, isolation of Lin^−^Ly6g^−^CD11B^−/low^Ly6c^int/+^CD115^+^c-Kit^+^F4/80^low^ cells from whole bone marrow may reflect the resident osteoclast progenitors derived from murine sources. CD45^−/low^CD115^+^ myeloid cells with co-expression of Ly6C^high^ or CCR2^high^, from blood likewise show robust CSF1/RANKL-dependent osteoclastogenesis, but further addition of Cx3cr1 may help to refine selection of osteoclast progenitors from peripheral blood.

## 4. OCP Changes During Disease, Injury, and Age-Associated Bone Loss

### 4.1. Changes to Osteoclast Progenitors with Advanced Age

Osteoclast progenitors both increase in number and have higher intrinsic osteoclastogenic potential with advanced age [88,89,90,91]. In a study using CD14^+^ cells isolated from the peripheral blood of women aged 49–66, age and menopausal status was shown to be positively associated with increased bone resorption in vitro [92]. Moreover, an epigenetic analysis of peripheral CD14^+^ cells suggested that increasing years post-menopause resulted in diminished DNA methylation of gene promoters of DC-STAMP and CTSK, genes associated with osteoclastogenesis and resorption, respectively [92]. In addition, circulating OCPs were found to have greater osteoclastogenic potential than other myeloid cells and that the frequency and activity of osteoclast progenitors negatively correlates with bone mass in post-menopausal women [93,94]. Moreover, estrogen suppresses the number of circulating OCPs in post-menopausal women [95].

Many studies demonstrate that the number of osteoclast progenitors increases with advanced age, both in pre-clinical models as well as in humans, along with age-related bone loss. With age, the composition of bone marrow shifts to diminished proportions of adaptive immune cells, increased adiposity, and enhanced myelopoiesis [88,89]. The expansion of myeloid cells with age includes myeloid-derived suppressor cells (MDSCs) that can serve as osteoclast progenitors [96,97]. Further analyses demonstrated that MDSCs were expanded within peripheral tissues, but not within the bone marrow, of aged mice [98]. In contrast, other studies demonstrate increased numbers of monocyte/macrophage cells within the bone marrow of aged mice [99]. This suggests that both bone marrow-resident and peripheral osteoclast progenitors expand with advanced age.

Other studies suggest that the mesenchymal compartment of the bone marrow supports myeloid skewing [100]. For instance, clearing of p16-expressing senescent osteoclast progenitors did not influence bone mass in a pre-clinical model, suggesting that other cell types, most notably osteocytes, may be responsible for age-related declines within the skeleton [90]. Changes in skeletal stem cells with advanced age were likewise shown to support an inflammatory microenvironment within the bone marrow that led to myeloid skewing and reduced bone regeneration [101]. Overall, studies support that osteoclast progenitor number and osteoclastogenic potential increase with age, but the underlying causes for this are still under study.

### 4.2. Osteoclast Progenitors Participating in Bone Healing Following Injury

Proper bone healing following injury requires the actions of osteoclasts. During endochondral fracture healing, osteoclasts aid in bone healing as part of the early and late phases [102,103]. This is likewise true of intramembranous bone healing of cortical defects [68], but clearance of osteoclasts via OPG treatment did not limit cortical bone defect healing; however, this conclusion was based on histological assessment alone at a single time point (e.g., 9 days post defect generation). Laser ablation studies likewise show that osteoblast injury precipitates an inflammatory response leading to the recruitment of immune cells and the formation of cathepsin K^+^ cells in zebrafish, with anti-inflammatory treatment limiting osteoblast recovery following injury [104].

There are several origins of osteoclasts that participate during bone healing. Parabiosis studies coupled with flow cytometry analyses in mice support that circulating Cx3cr1^+^ osteoclast progenitors participate in endochondral fracture healing [39]. In a separate study, parabiosis experiments coupled with single cell sequencing analyses also demonstrated that Cx3cr1 marks a population of erythromyeloid progenitors within the spleen that can be mobilized to participate in bone healing following injury [36]. Studies from a fracture healing model using medaka suggested that two populations of osteoclasts participate in early and late phases of bone healing, with the presence of a sealing or clear zone defining osteoclasts within the late phase of healing [103]. This observation suggests heterogeneity during the bone healing process and the potential for more than one progenitor source during different phases of healing [103].

Osteoclast progenitors participate in bone healing not only through the formation of bone-resorbing osteoclasts, but may also promote osteogenesis required for ossification following injury. In a study focusing on TRAP^+^ lineage cells, osteoclast progenitor cells were suggested to promote neovascularization and bone formation via PDGF-BB in a pre-clinical model of osteoporosis [105]; however, the exact cell type mediating this effect is unclear. Other studies identified periosteal osteoclast precursors promoting fracture healing through PDGF-BB in a cathepsin K-independent manner [106].

Osteoclasts not only participate during bone healing, but may also be needed for limb regeneration as well as integration of dental and orthopedic implants. Studies using a model of natural limb regeneration in axolotl demonstrate that osteoclasts prime the skeleton for successful regeneration and blocking their function limits limb regeneration [107]. Pre-clinical evidence also supports that integration of dental and orthopedic implants likewise requires the action of osteoclasts to aid in the resorption of necrotic bone, initiate bone formation at the implant interface, as well as during the late phases of implant integration [108,109,110,111,112,113]. Conversely, osteoclasts also contribute to implant loosening associated with debris wear particle production [114,115]. The developmental origins and sources of osteoclast progenitors mediating these processes are areas for further research.

### 4.3. Osteoclast Progenitors in Rheumatoid Arthritis

Whereas adult osteoclasts are predominantly HSC-derived and sourced from bone-marrow–resident progenitors during homeostasis, rheumatoid arthritis recruits osteoclast progenitors from additional sources resulting in the destruction of subchondral bone. This includes HSC-derived cells from the periarticular bone marrow, circulation, and spleen [53,74,77,116,117,118]. Expansion of osteoclast progenitors defined by surface marker expression of CD11B^−/low^CD117^+^CD115^+^ or high levels of CCR2 within the periarticular bone marrow space contributes to bone erosion associated with RA [53,77,119]. Studies using human peripheral blood also demonstrate that circulating osteoclast progenitors defined by CD11B/CD14 also expand with RA [120]. This is further corroborated by studies in mice demonstrating that peripheral and splenic OCPs expand in models of RA and that adoptive transfer of OPCs derived from the spleen hone to RA sites through a CCL2/CCR2-dependent mechanism [53]. Moreover, high levels of circulating TNF-α lead to expansion of circulating osteoclast progenitors [121], which may serve as a source of RANKL-independent osteoclastogenesis via a TGF-β-dependent mechanism [122]. In addition to bone marrow-resident and circulating OCPs, macrophages within synovium may also serve as a source of osteoclast progenitors. These arthritis-associated osteoclastogenic macrophages (AtoMs) are present within the inflamed synovium and intravital imaging in mice demonstrates their differentiation in situ to TRAP^+^ cells [116,117,118]. Despite these advances in our understanding of osteoclast progenitors that contribute to bone erosion in the setting of rheumatoid arthritis, studies largely lack definitive lineage tracing to quantify contributions of specific progenitors. Likewise, human tissue-level in vivo evidence of specific OCP subsets forming erosive osteoclasts remains limited.

### 4.4. Osteoclast Progenitors in Periodontitis

Increased osteoclast activity contributes to bone loss associated with periodontal disease. There are no definitive studies that demonstrate the developmental origins that contribute to osteoclast progenitors in the context of periodontal disease. In contrast, CD11B^+^c-Fms^+^Ly6C^high^ cells within the bone marrow, spleen, and peripheral expansion in response to chronic *Porphyromonas gingivalis* exposure implicate cells of hematopoietic origin [75,123]. Moreover, increased levels of CCl2 within the gingival crevicular fluid are associated with periodontitis, suggesting recruitment of osteoclast progenitors from the periphery [124]. Numerous studies also demonstrate that peripheral blood monocytes from periodontitis patients have enhanced osteoclastogenic capacity [125,126,127,128,129]. We discuss below how immune priming can enhance bone resorption in case of chronic periodontitis.

### 4.5. Impacts of Trained Immunity on Osteoclast Progenitors

Bone loss associated with chronic inflammatory diseases, such as periodontitis and arthritis, is driven in part by dysregulated osteoclastogenesis initiated in the innate immune compartment. A growing body of evidence implicates trained immunity—a memory-like state of the innate immune system induced by microbial components such as lipopolysaccharide (LPS), β-glucan, and live or inactivated pathogens—in the functional reprogramming of osteoclast progenitors (OCPs). This reprogramming includes persistent alterations in surface marker expression, metabolic activity, epigenetic profile, and bone-resorptive potential.

Studies using pre-clinical models have elucidated critical aspects of this relationship. Repeated or chronic exposure to the periodontal pathogen *Porphyromonas gingivalis* has been shown to expand a CD11B^+^c-Fms^+^Ly6C^high^ population with enhanced osteoclastogenic activity within bone marrow, spleen, and blood [123]. These inflammatory OCPs also exhibit altered transcriptional profiles related to differentiation, inflammation, and apoptosis [75,76]. Similarly, in arthritis models, disease-associated expansion of CCR2^high^ OCPs was observed, with transcriptomic profiling revealing osteoclast-priming signatures enriched in pathways such as chemokine and NOD-like receptor signaling [53]. Studies in the context of collagen-induced arthritis further revealed enhanced migratory capacity and osteoclastogenic potential of CD11B^+^CD115^+^ and CD11B^low^CD115^+^ subsets under inflammatory conditions, with significant changes in chemokine receptor expression [77].

At the mechanistic level, trained immunity has been causally linked to histone modifications (e.g., H3K4me3, H3K27ac) and metabolic shifts, such as increased glycolysis and TCA cycle flux, among myeloid progenitors exposed to β-glucan or Bacille Calmette–Guérin (BCG) [78,79]. These alterations confer long-lasting amplification of inflammatory outputs and cellular responsiveness. In vitro systems modeling trained macrophage-mediated osteoclastogenesis confirm that immune memory alters cytokine production (e.g., TNF-α, IL-6) and influences osteoclast differentiation in co-cultures [80,130]. Key signaling pathways included TNF-α/S100A8/A9, MyD88/NF-κB, and mTOR, all amplifying responses to RANKL and inflammatory cues [130,131]. Longitudinal in vivo analysis also suggests β-glucan priming of monocytes can extend functional changes for weeks, partially through survival enhancement and transcriptional reprogramming [132].

Taken together, these studies converge on the concept that trained immunity, as induced by microbial or inflammatory stimuli, permanently reshapes the osteoclast precursor landscape through both phenotypic transformation and molecular remodeling. Chronic inflammation reinforces osteoclastogenic potential at the progenitor level, providing a mechanistic link between innate immune memory and sustained bone loss in inflammatory diseases. Despite these advancements, further research is needed to establish the temporal persistence of trained immunity and the cumulative effects during aging or recurrent inflammation.

### 4.6. Osteoclast Progenitors in Osteoarthritis (OA)

Subchondral bone remodels in a time-dependent fashion during osteoarthritis disease progression. This includes diminished subchondral bone mass early during OA as well as enhanced or sclerotic bone late during joint degeneration [133]. Roles for bone resorption in OA progression are supported by studies demonstrating that anti-resorptive agents have effects on OA progression [134]. Current evidence suggests that osteoclast progenitors may contribute to OA progression by disrupting the osteochondral junction [135,136,137,138]. Human evidence supports that vascular channels traverse into cartilage in OA specimens [136]. Further studies in mice document that increased osteoclast progenitor-produced PDGF-BB drives type H vessel growth into the osteochondral junction [135,137,138]. To date, there is no definitive evidence that defines the developmental or anatomical origins of osteoclast progenitors that facilitate altered subchondral bone remodeling associated with OA. Recruitment routes of osteoclast progenitors to subchondral bone remodeling sites is likewise still under investigation. These are all areas for future research that would greatly enhance our understanding of OA pathogenesis.

## 5. Conclusions

In this narrative review, we provided an overview of the developmental sources, defining surface markers, and changes to osteoclast progenitors that occur in various conditions. While we aim to be comprehensive, our review of the field has several limitations. We acknowledge that various cancer cell types, including lung, breast and prostate cancer, frequently metastasize to bone and promote osteolysis. Other cancer types, including oral squamous cell carcinoma, osteosarcoma and multiple myeloma, are also characterized by altered bone remodeling. Offering a comprehensive review of how osteoclast progenitors change in each of these specific cancers is beyond the scope of this review. Our narrative review is also a biased summary of the field and is not comprehensive of all literature within the field that would be achieved by performing systemic review on a narrower research topic. Our understanding of osteoclast progenitors is limited due to a lack of unique markers, heterogeneity, limited numbers in vivo, and difficulty in studying these cells within their native niche(s). Combined approaches utilizing single cell sequencing, single cell epigenetics, and spatial transcriptomics could help to address these restrictions.

While we have made large advances in our understanding and treatment of conditions characterized by changes in the bone remodeling process, understanding how different osteoclast progenitors contribute to normal and pathological bone loss is still a key area for future research. Defining specific OCP surface markers that are associated with different disease contexts and conditions will allow us to understand how OCPs differ as compared to those involved in physiological bone remodeling. This could further aid the characterization of OCP types and development of tailored therapeutics.

## Figures and Tables

**Figure 1 ijms-26-10619-f001:**
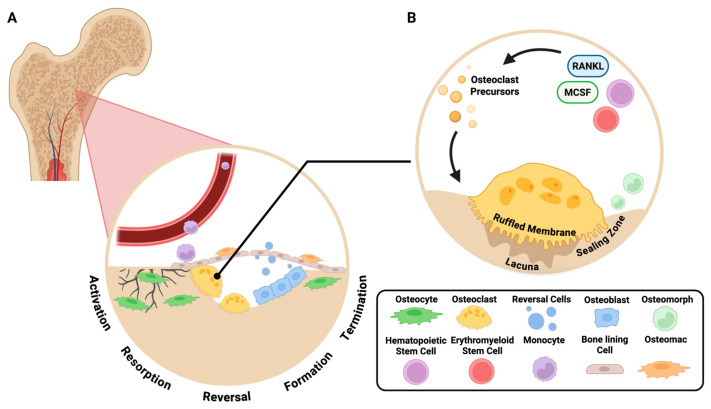
Osteoclasts play a key role in the bone remodeling cycle. (**A**) Bone remodeling is a constant cycle of activation, resorption, reversal, formation and termination facilitated by a multitude of cell types. (**B**) Osteoclasts derive from macrophage lineage cells and further differentiate into osteoclast precursors in response to RANKL and CSF1. Osteoclasts attach to bone surfaces and form a sealing zone in which the ruffled border acts to dissolve mineralized bone. Fission of osteoclasts further results in the formation of osteomorphs. Arrows denote progression through phases of osteoclast differentiation. *Created in BioRender. Vu, E. and Bradley, E. (2025) https://BioRender.com/pjfixc4* (accessed on 29 September 2025).

**Figure 2 ijms-26-10619-f002:**
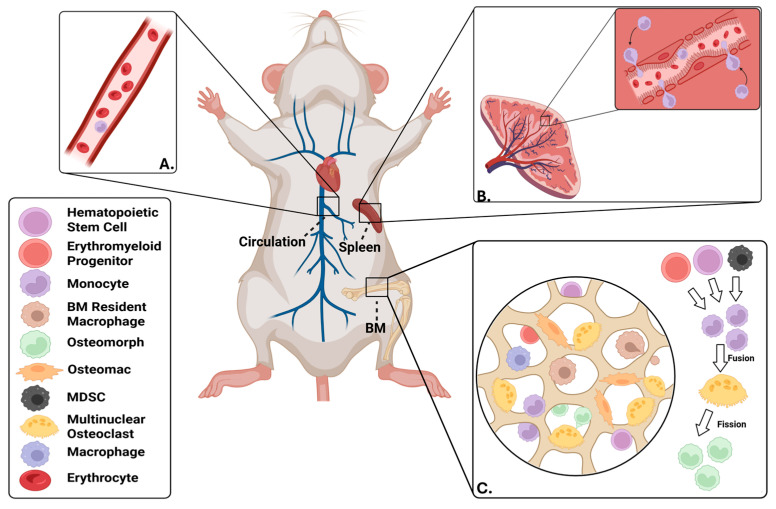
Sources of osteoclast progenitors. (**A**) Circulating CD14^+^ and CCR2^high^ monocyte with the ability to differentiate into multinucleated osteoclasts after chemokine-mediated migration to inflamed joints. (**B**) Mobilized splenic OCP after IL-34-induced differentiation in the splenic red pulp. (**C**) Heterogeneous, medullary bone environment demonstrating Hematopoietic Stem Cells, Erythromyeloid Progenitors, and Myeloid-Derived Suppressor Cells participate in monocyte/macrophage lineage commitment. Subsequent fusion of monocytes into multinucleated osteoclasts. Fission of the multinucleated osteoclasts demonstrating osteomorph formation. *Created in BioRender. Shimak, M. and Bradley, E. (2025) https://BioRender.com/pi7qhbh* (accessed on 29 September 2025).

**Figure 3 ijms-26-10619-f003:**
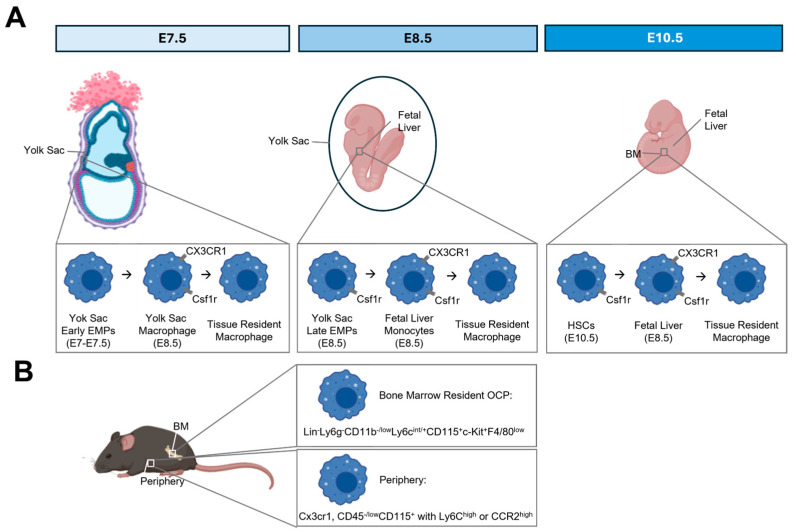
Surface markers for osteoclast progenitors at different developmental time points. (**A**) Osteoclast progenitors and respective surface markers during embryonic development. (**B**) Osteoclast progenitors and putative surface markers for adult bone marrow-resident and peripheral cells. *Created in BioRender. Bradley, E. (2025) https://BioRender.com/fuqmkg0* (accessed on 29 September 2025).

## Data Availability

No new data were created or analyzed in this study. Data sharing is not applicable to this article.

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
