# Peer review of "From Development, Disease, and Decline: A Review of What Defines an Osteoclast Progenitor"

_ijms, 2025, doi:10.3390/ijms262110619_

Round 1
Reviewer 1 Report
Comments and Suggestions for Authors
The authors of the manuscript entitled “From Development, Disease, and Degeneration: What Defines an Osteoclast Progenitor?” aim to summarize what is known about the developmental origins, 16 anatomical sources, and markers of osteoclast progenitors. The authors explored evidence related to the main sources of osteoclasts in both animals and humans and contrasted the relevant information regarding the surface markers that may define what an osteoclast is. The overall results are interesting to the field, and this peer reviewer encourages the authors to improve their manuscript by considering the following:
Major concerns:
- The overall design of this review is adequate; however, there is a lack of a clear path of information which is recommended to be further revised by the authors.
- In the introduction, it is recommended to use a figure to explain the bone remodelling process to understand key participants and concepts such as the bone remodelling unit.
- In the Figure 1 there is, apparently, a mistake in the close-up window of the spleen.
- Line 184: Please confirm if the intended term is “Intravital” instead “Intraviral”.
- Before determining surface markers for an osteoclast, it is recommended to use a detailed figure of what an osteoclast is, including (perhaps) histological images that may add to the morphological recognition of specific zones (such as the sealed zone).
- If the authors aim to add knowledge for the development of therapeutic strategies, there is a need for the description of molecular pathways activated by the surface markers involved in osteoclastogenesis and osteoclast-related bone resorption activities.
- In Chapter 4, age is discussed as a disease. It is suggested to revise the tittle of the chapter in order to classify correctly a condition such as age.
- In Chapter 4, the information is limited considering that osteoclastogenesis is different base on the condition/disease; therefore, the authors should revise how many conditions/diseases may represent the different pathways involved in osteoclastogenesis and how they are characterized. Importantly, consider diseases such as osteoarthritis, periodontitis, and bone-related cancer in further detailed.
- Discussion is too short and maybe it should be considered more as a take-home message, or a final statement of what is missing in the current knowledge.
- Overall, it is recommended to revise the tittle of the manuscript based on the content, as well as the type of the study (this manuscript can be considered a mini-review, but the authors should state why to make it this way – e.g., maybe as an up-to-date review of the last X years -).
Please revise some singular/plural mistakes, and grammar in general. Overall, English Language is readable.
Author Response
The authors of the manuscript entitled “From Development, Disease, and Degeneration: What Defines an Osteoclast Progenitor?” aim to summarize what is known about the developmental origins, 16 anatomical sources, and markers of osteoclast progenitors. The authors explored evidence related to the main sources of osteoclasts in both animals and humans and contrasted the relevant information regarding the surface markers that may define what an osteoclast is. The overall results are interesting to the field, and this peer reviewer encourages the authors to improve their manuscript by considering the following:
Major concerns:
Comment 1: The overall design of this review is adequate; however, there is a lack of a clear path of information which is recommended to be further revised by the authors.
Response 1: We made revisions to the text as suggested.
Comment 2: In the introduction, it is recommended to use a figure to explain the bone remodelling process to understand key participants and concepts such as the bone remodelling unit.
Response 2: We added Figure 1 (page 2) that depicts the bone remodeling process as well as basic osteoclast cell biology.
Comment 3: In the Figure 1 there is, apparently, a mistake in the close-up window of the spleen.
Response 3: We made this correction to current Figure 2 (page 4).
Comment 4: Line 184: Please confirm if the intended term is “Intravital” instead “Intraviral”.
Response 4: We corrected to “intravital.” Thanks! (Page 5, Line 196 of tracked changes manuscript version)
Comment 5: Before determining surface markers for an osteoclast, it is recommended to use a detailed figure of what an osteoclast is, including (perhaps) histological images that may add to the morphological recognition of specific zones (such as the sealed zone).
Response 5: Thank you for this suggestion. We added this information to Figure 1 (page 2).
Comment 6: If the authors aim to add knowledge for the development of therapeutic strategies, there is a need for the description of molecular pathways activated by the surface markers involved in osteoclastogenesis and osteoclast-related bone resorption activities.
Response 6: We added description of the pathways induced by each surface marker within the corresponding text (Lines 263-266, 283-284, 294 in tracked changes manuscript version).
Comment 7: In Chapter 4, age is discussed as a disease. It is suggested to revise the tittle of the chapter in order to classify correctly a condition such as age.
Response 7: We changed this subtitle to: OCP Changes during Disease, Injury, and Age-Associated Bone Loss (Page 7, Line 331 in tracked changes manuscript version).
Comment 8: In Chapter 4, the information is limited considering that osteoclastogenesis is different base on the condition/disease; therefore, the authors should revise how many conditions/diseases may represent the different pathways involved in osteoclastogenesis and how they are characterized. Importantly, consider diseases such as osteoarthritis, periodontitis, and bone-related cancer in further detailed.
Response 8: With the exception of bone-related cancer, we added discussion of OCPs within these disease contexts as well. We noted that lack of review of how the varied bone-related cancer changes osteoclast progenitors is a limitation of our manuscript, but it out of scope as an entire review article could be devoted to this question (Page 10 Lines 426-273, Page 13 Lines 540-554, Page 13 Lines 559-563 in tracked changes manuscript version).
Comment 9: Discussion is too short and maybe it should be considered more as a take-home message, or a final statement of what is missing in the current knowledge.
Response 9: We changed this title heading to “Conclusions”, as we discuss much of the relevant research within the body of the text (Page 13 Line 553 in tracked changes manuscript version).
Comment 10: Overall, it is recommended to revise the tittle of the manuscript based on the content, as well as the type of the study (this manuscript can be considered a mini-review, but the authors should state why to make it this way – e.g., maybe as an up-to-date review of the last X years -).
Response 10: We updated the title to reflect the article type.
Reviewer 2 Report
Comments and Suggestions for Authors
This review article submitted to IJMS by MDPI, titled “From Development, Disease, and Degeneration: What Defines an Osteoclast Progenitor?” by Shimak et al., 2025
- In the current research the authors addressed this is a very strong, high-level topic. The topic is original and relevant to the j and the field
Minor corrections:
- The abstract needs more details and better to be structured,
- keywords: need more words
- Several sentences are long and without ref.
- Please make sure the introduction to mention the entire biological continuum — from normal development to pathological states — suggesting a comprehensive, integrative perspective in the aim.
- Please split long sentences and add a ref. for each single sentence or each single info.
- To the short aim please better write the search strategy (SS)
- Add future directions
Major corrections:
- Please, in the introduction define “an osteoclast progenitor”,
- Also, define the mechanistic exploration of identity, lineage tracing, or cell heterogeneity,
- Address in the discussion the following under subheadings: “disease (e.g., osteoporosis, arthritis, cancer metastasis), and degeneration (aging),”
- However, some flawless are there to be difned better:
scope may be too broad—covering development, disease, and degeneration might make it hard to achieve depth and focus in one paper which need to be clearly framed.
- No Direct Mention of Methodological or Conceptual Framework:
- Add “Tracing the Identity of Osteoclast Progenitors from Development to Degeneration”
- No “mechanistic transformation and disease linkage”?!
- What about stem cells, please add info about this in details,
- More tables are required to comprehensively summarize the infor in the text,
- Figures, with no addition to the knowledge, these are known and shown in previous papers, please innovate and add new info to each and better to be summary to the text, enrich all figures more, they are shallow,
- Therapeutic strategies and all coming title heading needs subheading for each medication or each info below in other sections,
- Please organize btter
- Resistance mechanisms in needs subheadings for the info below,
- Enumerate the prognosis indices ?
- No conclusion or it is the discussion?
- List of abbreviations to be added
Author Response
Minor corrections:
Comment 1: The abstract needs more details and better to be structured
Response 1: We added detail to the abstract (Page 1 Lines 18-24 in tracked changes manuscript version).
Comment 2: keywords: need more words
Response 2: Additional keywords are included (Page 1 Line 26 in tracked changes manuscript version)
Comment 3: Several sentences are long and without ref.
Response 3: We made these corrections
Comment 4: Please make sure the introduction to mention the entire biological continuum — from normal development to pathological states — suggesting a comprehensive, integrative perspective in the aim.
Response 4: We include this within the introduction (Page 3 Lines 125-130 in tracked changes manuscript version).
Comment 5: Please split long sentences and add a ref. for each single sentence or each single info.
Response 5: We made these corrections throughout the manuscript.
Comment 6: To the short aim please better write the search strategy (SS)
Response 6: As this is a narrative review, we did not employ a defined search strategy as would be used within a systematic review.
Comment 7: Add future directions
Response 7: We note throughout the manuscript where there are discrepancies within the field, gaps in knowledge, or areas for further development. We also added more discussion within the conclusion section.
Comment 8: Please, in the introduction define “an osteoclast progenitor”,
Response 8: We made this correction within the introduction (Line 77 in tracked changes manuscript version).
Comment 9: Also, define the mechanistic exploration of identity, lineage tracing, or cell heterogeneity,
Response 9: Our manuscript summarizes key lineage tracing results that help to define the origins of different osteoclast progenitors. We also note where there are limitations to the current state of knowledge within our review of the literature.
Comment 10: Address in the discussion the following under subheadings: “disease (e.g., osteoporosis, arthritis, cancer metastasis), and degeneration (aging),”
Response 10: We added more discussion within the conclusions section, but also note that we discuss findings throughout our review (Page 13 Lines 556-596 in tracked changes manuscript version).
Comment 11: However, some flawless are there to be difned better:
scope may be too broad—covering development, disease, and degeneration might make it hard to achieve depth and focus in one paper which need to be clearly framed.
Response 11: We disagree as we are focusing solely on osteoclast progenitors.
Comment 12: No Direct Mention of Methodological or Conceptual Framework:
Response 12: We note that this is a narrative review. It is a reflection of the authors’ perspective does that does not utilize a structured methodology. We do describe the focus of this literature review within the abstract and introduction.
Comment 13: Add “Tracing the Identity of Osteoclast Progenitors from Development to Degeneration”
Response 13: This comment is unclear, and we are unsure to address the Reviewers critique.
Comment 14: No “mechanistic transformation and disease linkage”?!
Response 14: Unfortunately, we are unsure what the Reviewer is requesting with the comment. This is a critical review of the existing literature and we provide relevance to several pertinent conditions affecting bone mass (e.g., age-related bone loss, RA, periodontal disease).
Comment 15: What about stem cells, please add info about this in details,
Response 15: Within the introduction, we note that osteoclasts derive from hematopoietic stem cells and EMPs and discuss how these two developmental sources influence bone remodeling in different contexts.
Comment 16: More tables are required to comprehensively summarize the infor in the text,
Response 16: We added an additional Figure (Page 2).
Comment 17: Figures, with no addition to the knowledge, these are known and shown in previous papers, please innovate and add new info to each and better to be summary to the text, enrich all figures more, they are shallow,
Response 17: Our goal of this review is to summarize and critically review existing literature, pointing to areas where additional research is needed. We do this throughout the manuscript. We also added detail to Figures and note that Figure 3 summarizes “putative” surface markers for the isolation of OCPs. This panel of markers has not been utilized to isolate OCPs to date, so this marks an innovation within the manuscript.
Comment 18: Therapeutic strategies and all coming title heading needs subheading for each medication or each info below in other sections,
Comment 18: We are unsure what the Reviewer is requesting with this comment. We do not discuss medications and therapeutic strategies within our review. This is out of scope of our narrative review.
Comment 19: Please organize btter Resistance mechanisms in needs subheadings for the info below,
Comment 19: This comment is unclear, and we are unsure to address the Reviewers critique.
Comment 20: Enumerate the prognosis indices ?
Response 20: This comment is unclear, and we are unsure to address the Reviewers critique.
Comment 21: No conclusion or it is the discussion?
Response 21: We discuss most of the major findings within the field, noting gaps in knowledge and areas for further development within the texts. We changed the heading of this section to Conclusions.
Reviewer 3 Report
Comments and Suggestions for Authors
The problem of osteoporosis is urgent due to its higher prevalence, both among men and women, leading to fragile bones. The role of osteoblasts and osteoclasts in osteoporosis pathology, as well as the balance of their combined activity still remains an open issue.
The review has high scientific value and gives an outlook of osteoclasts role in the pathogenesis of osteoporosis. The description of the role of newly discovered osteomorphs and osteomacs in reversal phase of osteogenesis has a great interest; the activity of these osteoprogenitors opens new frontiers in the treatment of fractures and bone loss. Still, there are some notes which the reviewer had to clarify.
Page 2, lines 75-76. Perhaps, the second phase might be the resorption phase (see the line 80). Because the whole process is already known as remodeling, as the authors correctly point.
Page 2, line 88. A good idea would be to describe briefly osteoprogenitor activity of some biomaterials, especially when combined with growth factors. Though it is not the issue of this article, but the discussion of the role of bone-replacing materials in reversal phase might be significant to emphasize the niche role in osteogenesis.
Page 4, line 142. The acronym needs to be placed after ‘erythromyeloid progenitors’, because it is used further.
Page 10, line 450. A good idea would be to clarify the limitations of the review. Despite the fact, that the article is comprehensive, this postulation looks more positive and encourages the readers looking for additional information about other publications towards the osteogenesis.
Page 10, line 450. The Discussion section seems to be some concise. I think, it can be named as Conclusion, because the problematics described earlier in the text is a discussion itself. Also, this section can be expanded in the part of key findings (brief outlook of osteoprogenitors’ source and non-canonical osteoprogenitors).
Author Response
The problem of osteoporosis is urgent due to its higher prevalence, both among men and women, leading to fragile bones. The role of osteoblasts and osteoclasts in osteoporosis pathology, as well as the balance of their combined activity still remains an open issue. The review has high scientific value and gives an outlook of osteoclasts role in the pathogenesis of osteoporosis. The description of the role of newly discovered osteomorphs and osteomacs in reversal phase of osteogenesis has a great interest; the activity of these osteoprogenitors opens new frontiers in the treatment of fractures and bone loss. Still, there are some notes which the reviewer had to clarify.
Comment 1: Page 2, lines 75-76. Perhaps, the second phase might be the resorption phase (see the line 80). Because the whole process is already known as remodeling, as the authors correctly point.
Response 1: Thank you for pointing out this mistake! We made this correction (Page 3 Line 89 in the tracked changes manuscript version).
Comment 2: Page 2, line 88. A good idea would be to describe briefly osteoprogenitor activity of some biomaterials, especially when combined with growth factors. Though it is not the issue of this article, but the discussion of the role of bone-replacing materials in reversal phase might be significant to emphasize the niche role in osteogenesis.
Response 2: We added text to emphasize how our understanding of the reversal process could also improve use of biomaterials for bone regeneration (Page 3 Lines 102-103 in the tracked changes manuscript version).
Comment 3: Page 4, line 142. The acronym needs to be placed after ‘erythromyeloid progenitors’, because it is used further.
Response 3: We made this change as suggested (Page 3 Line 111 in the tracked changes manuscript version).
Comment 4: Page 10, line 450. A good idea would be to clarify the limitations of the review. Despite the fact, that the article is comprehensive, this postulation looks more positive and encourages the readers looking for additional information about other publications towards the osteogenesis.
Response 4: We added a description of the limitations of our review within the Conclusions (Page 13 Lines 556-569).
Comment 5: Page 10, line 450. The Discussion section seems to be some concise. I think, it can be named as Conclusion, because the problematics described earlier in the text is a discussion itself. Also, this section can be expanded in the part of key findings (brief outlook of osteoprogenitors’ source and non-canonical osteoprogenitors).
Response 5: We changed the heading to conclusions, added additional outlook as suggested (page 13 Lines 555-576).